# Aerosol Detection and Transmission of Porcine Reproductive and Respiratory Syndrome Virus (PRRSV): What Is the Evidence, and What Are the Knowledge Gaps?

**DOI:** 10.3390/v11080712

**Published:** 2019-08-03

**Authors:** Andréia Gonçalves Arruda, Steve Tousignant, Juan Sanhueza, Carles Vilalta, Zvonimir Poljak, Montserrat Torremorell, Carmen Alonso, Cesar A Corzo

**Affiliations:** 1Department of Preventive Veterinary Medicine, The Ohio State University, Columbus, OH 43215, USA; 2Boehringer Ingelheim Animal Health USA Inc., Duluth, GA 30096, USA; 3Department of Veterinary Population Medicine, University of Minnesota, St Paul, MN 55108, USA; 4Department of Population Medicine, University of Guelph, Guelph, ON N1G 2W1, Canada; 5Independent Swine Consultant, Barcelona 08195, Spain

**Keywords:** porcine reproductive and respiratory syndrome, porcine reproductive and respiratory syndrome virus (PRRSV), aerosol, airborne, transmission

## Abstract

In human and veterinary medicine, there have been multiple reports of pathogens being airborne under experimental and field conditions, highlighting the importance of this transmission route. These studies shed light on different aspects related to airborne transmission such as the capability of pathogens becoming airborne, the ability of pathogens to remain infectious while airborne, the role played by environmental conditions in pathogen dissemination, and pathogen strain as an interfering factor in airborne transmission. Data showing that airborne pathogens originating from an infectious individual or population can infect susceptible hosts are scarce, especially under field conditions. Furthermore, even though disease outbreak investigations have generated important information identifying potential ports of entry of pathogens into populations, these investigations do not necessarily yield clear answers on mechanisms by which pathogens have been introduced into populations. In swine, the aerosol transmission route gained popularity during the late 1990’s as suspicions of airborne transmission of porcine reproductive and respiratory syndrome virus (PRRSV) were growing. Several studies were conducted within the last 15 years contributing to the understanding of this transmission route; however, questions still remain. This paper reviews the current knowledge and identifies knowledge gaps related to PRRSV airborne transmission.

## 1. Airborne Transmission of Viruses—Introduction and Definitions

Airborne transmission of infectious agents is an important topic in both human and animal medicine that has received great attention over the past years [1,2,3,4]. A general understanding, based on conclusions from the peer-reviewed literature, is that the role played by airborne transmission of pathogens in the epidemiology of infectious diseases is debatable, and likely to vary according to intrinsic and extrinsic factors. For instance, type of strain has been identified as an important intrinsic factor on the probability of a pathogen becoming airborne [5]. Additionally, environmental conditions such as ambient temperature and humidity [6,7] have been shown to be important extrinsic factors. Other factors including host species, host population size, and type of facility in which hosts are located may also play a role but have not been formally reported in the literature.

For the purpose of this review it is important to define the following terms: droplets, aerosols, airborne and aerosol transmission, local spread, and porcine reproductive and respiratory syndrome. 

Droplets is a general term that refers to particles generated and expelled by an infectious individual. These particles can be categorized according to size, with the smallest droplets being referred to as aerosols. Large droplets (>100 µm) are deposited fast due to gravitational forces whereas medium droplets ranging from 5 to 100 µm, and small droplets, or droplet nuclei (aerosols < 5 µm) may remain airborne for longer periods and are responsible for aerosol transmission per se [4].

Aerosols are defined as suspensions in air (or in a gas) of solid, or liquid particles small enough that they will remain airborne for a prolonged period of time because of their low dropping velocity. Aerosols settle very slowly in still air, and because of that they may be easily carried over long distances by turbulences and air currents [2]. Aerosols can be infectious when they contain viable pathogens [8]. It has been reported that aerosol transmission of diseases depends on physical variables related to the infectious particles such as particle size [9], quantities of pathogen emitted, rate of droplet desiccation, and environmental factors such as temperature and relative humidity [3]. 

Airborne transmission of pathogens occurs both directly through aerosols being inhaled and landing into the respiratory system and through contaminated objects where droplets have settled turning them into fomites; with those two not being mutually exclusive [3,4]. For the remainder of this review, aerosol transmission is defined as the passage of microorganisms directly from an infectious individual to a susceptible receiver through aerosols (animal-level definition), which are particles expelled through exhaling, sneezing or coughing; or as the transmission of PRRSV from herd to herd by a virus travelling through the air (herd-level definition) [10]. These particles vary in size and their ability to remain airborne for longer periods of time and may result in direct infection of a susceptible individual [3,8]. Local spread or area spread are other terms commonly mistakenly used to refer to aerosol transmission; but it is important to note that local spread and area spread may involve other modes of transmission including transmission via vectors and fomites (e.g., transport, manure pumping equipment, and tools).

There is an extensive body of work in human medicine that describes the challenges with demonstrating aerosol transmission. Even though many review and hypothesis-driven papers are available in regard to the possible role of aerosols in infectious disease transmission in humans and animals [2,11,12], definitive field data on the role of aerosol transmission are lacking and restricted to small acute outbreaks in human populations [13]. As an example, many authors attempted to understand the role of aerosol transmission in human influenza. However, conclusions still differ in regard to the role of aerosols versus contact, and versus large droplet transmission via contamination of inanimate objects (e.g., fomite); highlighting the challenge of conducting these studies [12,13,14,15].

Porcine reproductive and respiratory syndrome virus (PRRSV) is an RNA virus that infects swine of all ages. Two PRRSV species, PRRSV-1 (former genotype 1 or European-type) and PRRSV-2 (former genotype 2 or North American)-type, are currently recognized by the International Committee on Taxonomy of Viruses; but it is important to note that both virus species are found worldwide. PRRS is considered the costliest disease in the United States (US) swine industry. The virus produces respiratory and reproductive disease in pigs leading to poor performance and increased mortality [16]. Swine are the only known hosts for PRRSV [17] and can become infected by several routes of exposure including parenteral, intranasal, intramuscular, oral, intrauterine, and vaginal [18]. Once PRRSV is introduced into a population of susceptible animals, infection will occur, and infected animals will begin to shed the virus via saliva, nasal secretions, urine, semen, mammary secretions, and feces [19,20]. Numerous epidemiological studies have investigated farm-level risk factors for PRRSV outbreaks. Common significant factors from those studies include regional swine density, purchasing PRRSV positive semen or animals, and herd size [21,22,23,24,25]. Even though regional swine density continues to be hypothesized as one of the most important risk factors for PRRSV outbreaks, the reasons why this is so still require further understanding. It has been hypothesized that farms located in high dense areas are more prone to being exposed and infected with PRRSV through airborne transmission from infected surrounding sites. Transmission of PRRSV between farms at long distances via air continues to be investigated, and discordant results can be found in the literature supporting or opposing the idea of aerosol transmission between herds. 

The next sections of this review paper will examine peer-reviewed publications on the ability of PRRSV to become airborne, and its detection and transmission via air over different distances in order to identify evidence and knowledge gaps on the subject of aerosol transmission of PRRSV.

## 2. Characterization of PRRSV Aerosols

Few peer-reviewed publications have characterized aerosol particles containing PRRSV, even though it is recognized that particle characteristics are important for virus viability and therefore, transmission. Alonso et al. [9] challenged pigs with PRRSV in a controlled setting resembling farm conditions to characterize emissions. Aerosol particle concentration, size distribution, and infectivity of swine viruses including PRRSV were assessed. Authors reported that prevalence estimates varied according to the air sample collector used, with 23.5% and 5.1% of the samples testing positive for the cyclonic collector and the Anderson Cascade Impactor stages, respectively. Higher virus concentrations were found in larger particle sizes (e.g., 0.9–10 µm) with > 70% of those samples yielding a positive result on virus isolation in cell culture. These results support the fact that PRRSV can become airborne as well as remain infectious, especially in large particles.

Findings regarding the relatively low detection rate of PRRSV have also been reported with other similar enveloped swine viruses under field conditions. For example, the same research group [26] performed an experimental challenge study and reported that porcine epidemic diarrhea virus (PEDV) inoculated animals shed PEDV and the virus could be detected by RT-PCR in all air samples collected under experimental conditions inside a research facility. Furthermore, these researchers also collected samples from acute outbreaks under field conditions. Even though approximately 18% of those air samples tested RT-PCR positive (up to 10 miles), they were unable to infect naïve animals via gavage. This lack of infectivity was attributed to the lower viral concentration in field samples, inactivation of the virus by temperature, light intensity, ultra violet radiation, or sample storage.

Previous research has shown that PRRS viruses can survive in aerosols for varying amounts of time, and aerosols containing viable PRRSV can infect animals [27,28]. Specifically, for PRRSV, many other factors were shown to play an important role in virus survival and transmission via aerosol, such as temperature and humidity; with the virus being more stable at lower temperatures and/or lower relative humidity [29]. Virus’ capacity to become airborne has also been shown to vary by strain [5] and by median infectious dose [30]. These factors should be considered when evaluating and interpreting findings from available experimental or semi-experimental studies in this topic.

## 3. Aerosol Detection and Transmission of PRRSV

Peer-reviewed literature on the ability of PRRSV to become airborne and infectious has been available since the late 1990s and early 2000s as answers for unexplained PRRSV outbreaks were being sought. Knowledge generation started with studies at the experimental level that aimed at developing methods for airborne virus detection. Subsequent studies proceeded to confirm whether the virus could indeed become airborne and whether it had the capability to infect susceptible pigs. These studies were later followed by semi-experimental studies and finally, a limited number of field studies.

### 3.1. Studies Under Experimental Conditions

Torremorell et al. [27] was one of the first who published a peer-reviewed article to report short-distance airborne PRRSV transmission by documenting the seroconversion to PRRSV of a group of pigs housed in a chamber that were exposed to aerosols originating from PRRSV experimentally infected pigs housed in a separate chamber, but connected through a 1-m long pipe, confirming airborne transmission. This was also one of the first studies to document PRRSV strain differences (MN-1b and VR-2332) in transmitting PRRSV via air. Similarly, Wills et al. [19] used a PRRSV strain that was causing an acute outbreak in a herd (ATCC VR-2402) to conduct five trials using three raised nursery decks placed 48–102 cm apart in an isolation room. Infected pigs were placed in the middle of the room (center deck), indirect close-contact pigs were placed in one of the sides, and indirect contact pigs separated by a single sheet of aluminum were placed on the other side of the middle deck. Exposure lasted for 29 days. There was lack of evidence of transmission in two of the five trials for indirect close-contact pigs and lack of evidence of transmission in three of the five trials for indirect contact pigs separated by the aluminum sheet. Considering that animals were housed in the same physical space, authors concluded that airborne transmission may be less likely than anecdotally believed at the time. Another study was conducted in which pigs were housed in two separate environments but connected through pipes in which air flow was controlled. Researchers conducted this experiment by forcing air from PRRSV-infected pigs’ environment into the environment of naïve pigs using three different air exchange rates. In all three scenarios, investigators demonstrated that PRRSV could infect the naïve pigs [31]. 

Dee et al. [32] was one of the early research teams that aerosolized PRRSV MN-30100 and dispersed it using a cooking oil spritzer and a pump. The investigators assessed the probability of PRRSV becoming airborne and maintaining its infectiousness by aerosolizing the virus and sampling the air at different distances ranging between 1 and 150 m. Results from this experiment showed that PRRSV could be detected at all sampling distances and remained viable with log concentrations decreasing by 50% at the 33-m distance. This early work showed that PRRSV could remain airborne through distances of over 1-m. The question on whether these artificially generated aerosols would be different than animal-generated aerosols, and whether this would be important for transmission, still remains. It is important to mention that the authors recognized that large quantities of air with large amounts of virus had been inoculated for this experiment, which may not represent field conditions. This study also involved the exposure of six naïve pigs to PRRSV positive air, which resulted in 50% of those animals deemed to have been infected with the virus. Once more, the authors concluded that considering a large quantity of air had been inoculated with large quantities of PRRSV for this study, transmission of PRRSV by aerosols would be a rare event under field conditions, if occurring at all, and suggested that other established routes of infections should be prioritized [32].

A few years later, Cho et al. [5] was able to demonstrate that airborne transmissibility of PRRSV was strain dependent, suggesting that PRRSV pathogenicity was likely associated with the ability of the virus to become aerosolized. These authors used two PRRSV strains, a low pathogenicity strain (MN-30100) and a highly pathogenic isolate (MN-184), and a chamber model where one chamber was connected to a second chamber via a 1.3-m-long rectangular duct. The first chamber housed a group of 10 PRRSV positive pigs, whereas the second chamber contained an air sampling device. Virus shedding and clinical signs were assessed from all animals experimentally infected and the air sampling device was run three times for four days after inoculation as a means to assess whether the virus had become airborne. Virus was detected in the air in three out of five air samples that originated from animals infected with the PRRSV MN-184 strain whereas there were no positive results reported from samples from pigs inoculated with the PRRSV MN-30100 strain. Transmission of PRRSV by aerosol was detected in four out of ten replicates involving pigs inoculated with MN-184. In contrast, there was no evidence of aerosol transmission of PRRSV for animals inoculated with the MN-30100 PRRSV strain. In agreement with the above findings, PRRSV RNA levels in collected blood samples were significantly higher and clinical signs were more severe and of longer duration for animals experimentally infected with the MN-184 PRRSV strain compared to animals infected with the MN-30100 PRRSV strain.

### 3.2. Studies Under Semi-Experimental Conditions

For the purpose of this review, semi-experimental conditions refer to studies under a controlled environment but using a large population of animals. These studies are not conducted under controlled research conditions but using similar facilities to what would be encountered under field conditions. Studies assessing short-range PRRSV aerosol transmission (120 m or less) will be discussed first, followed by studies assessing long-range PRRSV aerosol transmission.

Otake et al. [33] exposed naïve pigs to 210 PRRSV MN-30100 infected pigs in a controlled field study by either housing them in different pens within the same barn as the infected pigs, or by housing naïve pigs in two trailers that were placed 1 and 30 m away from the exhaust fan of the barn housing the infected pigs. Animals remained in the trailers for 72 h and were moved to two other small facilities after this period (located at approximately 30 and 50 m from the facility containing infected animals), where they stayed for another 21 days. Air samples were collected four times daily for the 72-h period. Results from this experiment showed that indirect contact negative pigs inside the infected barn were able to become infected even when located over 1 m of distance. However, pigs housed in the outside trailers did not get infected even though there was evidence that the virus was circulating extensively in the infected barn and the study was conducted under weather conditions thought to support PRRSV survival. Furthermore, all collected air samples tested negative by PRRSV PCR. The authors concluded that aerosol transmission appears to be at best an infrequent event and suggested that outbreaks due to airborne PRRSV transmission should not be used as an excuse for inadequate investigation of other possible biosecurity risks [33]. 

Using a similar study design, Trincado et al. [34] attempted to transmit PRRSV MN-30100 via air under semi-experimental conditions by placing ten negative sentinel pigs in a trailer and exposing them to air from the exhaust fans of a building containing 150 experimentally infected finishing pigs for seven days using a 15 m plastic tube. This attempt to transmit the virus was not successful, even though both direct and indirect-contact controls located within the same air space became viremic during the exposure period. In addition, all of the 84 air samples collected from the infected barn and all of the 84 samples collected from the exhaust were negative by PCR, virus isolation, and pig bioassay. This work confirmed previous observations that highlighted the challenge of transmitting PRRSV experimentally by aerosols from one space to another, especially considering that under the conditions of this study, air was being directly transferred from an infected barn to a naïve trailer. 

In a similar study, Fano et al. [35] evaluated the air transmission of PRRSV MN 30100 in the presence of *Mycoplasma hyopneumoniae* (*M. hyopneumoniae*). Incidence of infection through direct animal contact, indirect contact between animals located in the same building and indirect contact by pigs located in a trailer placed in front of the exhaust fan coming from the barn containing infected animals was assessed. Results showed that all (12/12) of the direct contact animals became infected and that 60% (6/10) of the indirect contacts sharing the same environment but located 2.5 m away from the infected animals yielded a PRRSV RT-PCR positive result. None of the pigs housed in a trailer located in front of the exhaust fans (distances of 1 and 6 m) became infected. Similar to previous studies, these authors concluded PRRSV airborne transmission would be a rare event.

After this report, larger studies continued to attempt to resemble field conditions. Pitkin et al. [36] developed a swine production region model where one building with 300 PRRSV MN-184 infected pigs was located 120 m away from two other buildings containing 20 naïve pigs each. Incoming air on one building was filtered, whereas the other was not filtered. Despite PRRSV negative swabs of personnel and equipment, 31% (*n* = 8) of the 26 replicates of the study resulted in infection of the population housed in the non-filtered facility, suggesting a role of aerosol transmission. Additionally, 10.5% of air samples collected at the air inlet of such facility tested positive on virus titration. None of the replicates in the air filtered building became PRRSV positive. Twenty six percent of 190 aerosols samples collected outside the exhaust fan were positive using virus titration, even though there was robust evidence that the population was highly infectious. The authors also reported meteorological conditions that were associated with ability to detect virus in aerosols; however, it is important to note that air samples were only taken inside the buildings containing naïve animals. 

A similar attempt to detect the virus in aerosols has recently been made in Europe. Stein et al. [37] evaluated the ability to detect PRRSV-1 (European-type) in aerosolized samples in an experimental setup and under field conditions (nursery and finishing animals) using three different devices. It is important to note that most previous research had only investigated PRRSV-2, which is the North American PRRSV strain [38]. European researchers found that virus was recovered and detected by qRT-PCR in all the devices used to sample the air but only the sample collected during the higher PRRSV air concentration of a specific sampler yielded a positive result to virus isolation. They were not able to detect PRRSV-1 in any of the samples collected by the different air collectors when those were placed inside three different barns housing positive pigs. Based on those results, authors suggested the low level of viremia of the pigs in the barn could have been the cause of the failure in detecting the virus in the environment.

One of the first studies to evaluate long-distance transport of PRRSV via air was performed by Dee et al. [28]. In that study they experimentally inoculated 300 grow-finish pigs with PRRSV strain MN-184 and attempted to detect it via RT-PCR on air samples collected over a 50-day period over distances of 1.7, 2.6, 3.3, and 4.7 km from the infected herd in 16 designated points that included all four cardinal directions. From a total of 50 samples collected at the exhaust fan, 34% yielded positive results by RT-PCR, even though animals were actively shedding and transmitting the virus as evidenced by extensive animal sampling protocols. A total of 1.3% (*n* = 4) of 306 air samples collected in long distance were positive by RT-PCR and bioassay at the 4.7 km distance. A few points brought by this and other authors in regard to the difficulty in detecting PRRSV in the air under field conditions relate to whether animals shed enough amounts of virus when they are infected, if the external conditions may present challenges for virus travel and survivability, and whether it is possible to capture and detect viruses in air using our tools and protocols [28]. 

A complementary study [39] was performed to assess whether different PRRSV strains could be transported over longer distances ranging from 1.4 to 10.2 km and remain infectious in air samples. A population of approximately 252 grow-finish pigs that had been endemically infected with PRRSV strain 1-8-4 and *Mycoplasma hyopneumoniae* 232 was challenged with two heterologous PRRSV strains (PRRS RFLP [restriction fragment length polymorphism] 1-18-2 and 1-26-2) by the introduction of experimentally infected pigs. Air samples were collected from this source population and at 31 locations around the facility over a 21-day period, and bioassay was further conducted to determine viability of detected viruses. Out of 114 air samples collected around the infected population, five (4.4%) yielded a PRRSV positive RT-PCR result, further confirmed by ORF5 sequencing as the resident 1-8-4 strain; which was infectious from air samples by viral isolation and yielded positive results by bioassay. In contrast, long-distance airborne transport of PRRSV strains 1-18-2 and 1-26-2 was not supported by study results [39]. 

Finally, Linhares et al. [40] described the use of a PRRSV modified-live vaccine as a tool to reduce viral shedding to the environment, including aerosols. The study was conducted in a large research facility using two rooms of approximately 1000 growing animals each; from which 100 in each room were challenged at eight weeks of age with a PRRSV strain (RFLP 1-18-2) and from which one of the rooms received a modified-live vaccine at eight and 36 days post-infection (therapeutic vaccine use). Air samples were collected daily up to 118 days post-infection, and results showed that even though there was no difference detected in PRRSV RNA concentration in air samples between the vaccinated and non-vaccinated populations, the challenge-vaccine group had a shorter period of time in which air samples yielded RT-PCR positive results for PRRSV RNA [40]. This suggests an important point that underlying herd immunity may play a role in aerosol excretion, levels of detection, and potential for transmission to nearby swine populations. 

### 3.3. Studies Conducted Under Field Conditions

Large scale epidemiological field studies investigating the role of PRRSV airborne transmission as a cause of outbreaks are scarce, difficult to conduct, mostly retrospective in nature, and for the most part rely on diagnostic or monitoring systems already in place. 

Edwards [41] was one of the first authors to suggest the plausibility of airborne transmission between farms after several PRRSV outbreak investigations took place in the United Kingdom (UK), opening the research field for long-distance airborne transmission studies. In that early PRRSV investigation, researchers assessed inter-herd transmission of PRRSV after the disease was first detected. Even though 50% of the first 30 cases after the introduction of the virus in the country were linked to animal movement, that proportion went down to 18% after 100 cases in favor of local airborne spread (63% of the cases). Authors advocated that airborne spread was strongly suspected as animal movements were restricted in an effort to control and contain the disease. Based on the data and investigations, they claimed that airborne transmission should occur mostly over a 3 km radius from an infected herd. A few years later, Christianson and Joo (1994) summarized a scale from the Edwards et al. [41] study by calculating that approximately 57%, 31%, 11%, and 0% of herds around a PRRSV case were infected within 1 km, 1–2 km, 2–3 km or over 3 km, respectively [42]. It is important to note that at the time of the Edwards study [41], the UK pig population would have been naïve to PRRSV infection providing the best conditions for generating aerosolized PRRSV given the lack of immunity in the herds. 

In the early 2000s, Mortensen et al. [22] conducted extensive risk factor analysis using herd-level data from the first Denmark PRRSV outbreak due to the dissemination of PRRSV modified live vaccine virus. Using measures of local spread via “neighborhood exposure” variables that incorporated presence of neighbors, size of neighbors, and days of exposure, the authors concluded that aerosol transmission had likely been responsible for a great proportion of infected farms with biosecurity practices not being able to prevent herd infection. It is important to note this study was conducted under European field conditions (e.g., a European PRRSV genotype), and their investigation was focused on the spread of a US modified-live virus introduced after detection of the outbreak. This observation through a large epidemiological study stimulated a whole body of work trying to test such hypothesis via experimental trials, which were described in the previous sections of this review.

The first investigation of PRRSV genetic variability on a local geographical scale was conducted by Goldberg et al. [43] using swine sites located in the US states of Illinois and eastern Iowa. The authors hypothesized that a positive association between PRRSV genetic similarity and sites’ geographical proximity would support distance-limited processes as important for PRRSV spread such as the aerosol means of transmission. Fifty-five PRRSV isolates submitted to a diagnostic laboratory during 1997–1998 were sequenced using the ORF5 gene. The correlation between the proportions of differing nucleotides and the geographical distances between all pairs of sequences was investigated, taking into account days separating the submission as a potential confounder. Results showed there was a high level of genetic variability in the ORF5 gene of PRRSV isolates collected from this geographic area within the US, but that there was no significant association between genetic similarities and either geographic distance or temporal distance for the PRRSV isolates. These results suggested that movement of PRRSV isolates directly from farm to nearby farms could not explain the pattern in genetic variability found in this sample set, and that PRRSV may instead move via long-distance processes such as transport of animals or suppliers. In contrast, Mondaca-Fernandez et al. [44] conducted a similar study using data from a single production company located in the US state of Minnesota concluding that genetic similarity had a significant negative correlation with geographic distance.

Alonso et al. [45] investigated the potential for air filtration systems to protect herds from aerosol transmission of PRRSV by gathering data from 20 filtered and 17 non-filtered sow herds and comparing PRRS incidence rates over a seven year-study period. The study showed air filtration systems to be effective in reducing the frequency of PRRS outbreaks in sow herds and using data from their study the authors inferred that approximately 80% of new PRRSV introductions into herds with good biosecurity in swine dense regions may be attributable to the airborne route. As acknowledged by the authors; however, it is important to note that the decision and implementation of air filtration is an expensive investment that likely combines other biosecurity procedures. Therefore, quantifying the amount of reduction in PRRSV introduction due to these other biosecurity improvements versus the filtration system, per se, remains unknown. 

Rosendal et al. [10] used sequencing-based diagnostic testing (RFLP’s and ORF5 sequencing) from a diagnostic laboratory servicing swine sites across the Canadian province of Ontario over a three-year-period to investigate patterns of PRRSV distribution that could indicate area spread of PRRSV. Even though aerosol transmission is only one of several events that would result in spatiotemporal patterns in PRRSV genotype distribution, these authors captured confounders such as herd-level ownership and service suppliers (e.g., gilt source) in order to shed light on aerosol transmission per se. Their conclusion was that there was no strong evidence of aerosol transmission happening between pig herds in Ontario, Canada, according to the data examined. The evidence pointed instead towards transmission of PRRSV occurring in this population by common sources of animals and other herd inputs, except for one RFLP type (RFLP 1-3-4); whose cluster was interpreted as a possible example of area spread. Furthermore, this study highlights the importance of capturing herd networks when investigating PRRSV outbreaks. A follow-up Canadian study [46] using a different swine site population that consisted of swine sites participating in PRRS area regional control and elimination projects in the province of Ontario, Canada yielded similar results to Rosendal et al. [10]. The authors conducted clustering and hotspot analysis in order to investigate the presence of spatial clustering and clusters of PRRSV-positive sites located in three distinct regions. The conclusions were that clustering analysis was not able to identify that distance to positive sites plays a significant role in the PRRSV status of neighboring sites. A few years later, another follow-up study [47] that differentiated between three PRRSV strains (RFLPs 1-22-2, 1-3-2, and 1-8-4) and incorporated herd-level connectivity via transportation company information, reported that the importance of area spread and transportation network on PRRSV occurrence differed according to genotype.

Brito et al. [48] measured the frequency of detection and the quantity and diversity of PRRSV in daily air samples collected around four sow farms located in different pig density areas, from October to December 2012. Those authors concluded that the risk of the virus entering farms via aerosol was very high, with 37% (80/217) of air samples positive, with a minimum of 29% and a maximum of 42% depending on the farm. Furthermore, the authors reported diverse populations of PRRSV variants being present in the air, from eight to 14 different strains in one location using ORF5 sequencing. Similarly, a field study aiming at understanding frequency of exposure and diversity of PRRSV airborne viruses around filtered farms during the PRRSV high season yielded conflicting results compared to previous reports. Researchers collected a total of 241 air samples, using previously described methods, accounting for 482 hours’ worth of air sampling. None of these air samples yielded a positive RT-PCR result, which was unexpected especially considering that there were farms in the sampling area that were undergoing an outbreak of PRRSV [49].

Additionally, it has been suggested that the traffic of pig trucks would be a potential risk and source of PRRSV for farms located by highly transited swine-truck routes due to the aerosolization of virus and particles coming from the haulers. However, a study published as an abstract at a Conference Proceedings [50] attempted to elucidate the role of trucks on PRRSV aerosolization and potential aerosol transmission. Results showed that the percentage of positive air samples collected following a PRRSV positive pig truck was low [50]. In that study, 21 different loads of pigs from positive sources were followed by a car with a mounted cyclonic air collector on the roof for a duration of 15 minutes to two hours. Frontal parts of the car (hood, windshield, and bumper) were wiped and content was collected from the cyclonic collector after each trip. All 63 samples from collected car polyester pad swabs tested negative by PRRSV RT-PCR and only one (4.7%) out of the 21 cyclonic air collector samples yielded a RT-PCR PRRSV positive result; which was a borderline positive and deemed false positive. Authors of that study suggested that those results could be due to the age and level of shedding of the pigs.

In 2010, Dee described a study where 10 filtered and 26 non-filtered herds were followed for a period of 24 months [51]. Significantly fewer new PRRSV introductions were reported on filtered farms (0.2 cases per farm) than in non-filtered (1.4 cases per farm). The two reported filtered farm PRRSV introductions were attributed to contaminated transport, and breaches in biosecurity protocols. The authors concluded that aerosol transmission could not be completely eliminated as a route of entry but was less likely given the results of the outbreak investigation. 

In 2010, Spronk et al. [52] reported a similar study, where PRRSV introductions on two filtered, and five non-filtered herds over a 12-month period. Neither of the filtered herds reported a new PRRSV introduction, while all of the non-filtered herds reported a new PRRSV introduction. Additionally, air samples were collected outside one of the filtered farms during a 42-day period. Out of 73 air samples collected, two were found to be positive for PRRSV by RT-PCR; however, virus viability via bioassay or cell culture was not assessed. 

Recently, Arruda et al. [53] published one of the few prospective case studies involving field data that aimed at investigating the role of area spread in PRRSV transmission in swine dense regions in the U.S.; including the swine dense regions of North Carolina and Iowa. The authors took advantage of three naturally occurring sow farm PRRSV outbreaks to sample surrounding farms and check whether the virus identified on the focus outbreak farm could be found in neighboring farms. For none of the three scenarios the source virus was detected by ORF5 sequencing in any of the surrounding farms; which did not support the area spread theory as the main cause for these outbreaks. However, it is important to note that the ability of obtaining sequencing information from oral fluids for this study was low, from 20% to 33%, and the time range used for collection of samples was broad, within 60 days of the index outbreak.

A summary of the studies discussed herein is shown in Table 1 and Table 2.

## 4. Knowledge Gaps and Challenges with Aerosol Transmission Field Studies

This PRRSV aerosol transmission review has highlighted the body of available literature on this topic. More importantly, it has showed how challenging it is to conduct airborne detection and transmission studies as methods are not, at the present time, completely developed and fine-tuned to bridge the gap between experimental settings and field conditions. These challenges may help explain the conflicting results found in the literature with regards to this transmission route. The main challenge in designing real-life and scientifically sound studies to understand this issue relies on the fact that, in order to show the significance of a particular route, other potential confounding routes must be considered and controlled. As a result, designing experiments and drawing conclusions from epidemiological data represent a challenge under field conditions. Categories of evidence should be considered when attempting to determine the modes of transmission of a respiratory pathogen, which include survival of the pathogen in the environment, experimental infections under laboratory conditions, and epidemiological studies of naturally-occurring infections [12]. 

When faced with the dilemma of whether a PRRSV introduction was caused through the aerosol route or other means, it is important for swine veterinarians, producers, and educators to consider other modes of transmission that have been previously demonstrated and replicated for PRRSV by several peer-reviewed publications. These include direct [54,55] and indirect connections between swine sites [56], landscape and weather-related factors [29,33,57,58], site specific internal and external biosecurity measures [59,60], and other possible system-related commonalities between sites that are not always easy to measure but continue to play a role in transmission.

It is important to highlight in this review that the vast majority of experimental and semi-experimental studies conducted to elucidate the role of aerosol in PRRSV transmission were conducted in the US Midwestern region, which may represent a specific set of environmental conditions. Furthermore, studies on airborne transmission of PRRSV-1 species are scarce and contributions in this area will be valuable considering that PRRSV-1 and PRRSV-2 are substantially different viruses. Replication and development of the studies presented herein in other parts of the world and using PRRSV-1 (besides PRRSV-2) is warranted to continue uncovering characteristics of this transmission route. Unfortunately, airborne transmission or detection studies tend to be costly and it may be playing a limiting role for replication under other field-based conditions.

Additionally, data on outbreak investigations of PRRSV outbreaks is scarce. There are opportunities to indirectly elucidate the potential route of introduction of the virus, but current methods have not allowed to consistently draw sound conclusions, which then forces the investigation to prematurely default into aerosol transmission as the introduction pathway. Furthermore, current PRRSV regional sequence data has shown that virus diversity is important, which leads to think that if airborne transmission occurred even at a low rate, comparison of viruses at the regional level would yield high similarity results, but unfortunately that is not the case. It is, however, important to acknowledge that given the current lack of standard testing and reporting methods, this would have been difficult to capture and perhaps is inviting the industry to generate more PRRSV sequences in order to better assess the regional molecular epidemiology of this constant changing virus.

Furthermore, PRRSV airborne detection studies under field conditions are scarce. Most of these have been conducted with similar sampling devices, which may have limitations. In addition, conflicting results have been reported in studies using the same sampling device, warranting further investigations in order to clarify the frequency of airborne detection especially in high dense regions as a measure of risk. Future development of new air sampling methods may be helpful to more accurately investigate risk of aerosol transmission, and thus further our understanding of PRRSV aerobiology.

Another important factor to consider is the underlying immunity for current pig populations under commercial conditions. As previously stated, some of the strongest evidence of area spread comes from studies conducted in populations that were mostly susceptible before introduction of the virus [41], or populations with unknown level of exposure to a different PRRSV type [22]. In contrast, and especially as our understanding of PRRSV immunity management is progressing, many herds have consistently high level of immunity accomplished either via vaccination, live-virus inoculation, or a combination of both methods. This complicates the level of generalizability of the vast majority of available peer-reviewed research studies; which mostly utilize naïve animal populations. It is important to mention; however, that there have been examples of cases in which area spread was not deemed important during PRRSV introduction events with successful post-outbreak eliminations [61,62]; which highlights the fact that PRRS, as with other diseases, is multifactorial; and area spread is complex and not only agent and host-dependent; but also environment and area-dependent.

In general, aerosol transmission still requires further investigation as the process by which a virus originates from an infectious population and successfully infects a susceptible population is not well understood. Different probabilistic events need to occur for a successful transmission via air including: (1) large population of actively shedding pigs at a specific concentration, (2) probability of the virus becoming airborne, (3) probability of the virus exiting the pig barn in a viable state, (4) probability of the virus maintaining viability and high enough concentration while airborne as a dilution factor occurs, (5) probability that the virus reaching the neighboring farm and entering the building and finally, (6) probability that the infectious virus reaches target cells within a susceptible individual. 

Finally, the role of aerosols in contaminating the imminent environment via particle deposition has not been fully investigated for PRRSV. This has been shown for other viral diseases such as highly pathogenic avian Influenza [15] but more work is needed to fully understand the role of airborne spread at contaminating surfaces that then become source of PRRSV introduction through fomites into farms.

## 5. Conclusions

Overall, transmission of PRRSV continues to be an important area of research as with current knowledge and biosecurity procedures, herds continue to break highlighting the fact that there are still opportunities to further understand pathways of PRRSV introduction to herds. Several studies point towards a benefit of biosecurity measures in preventing virus introductions; however, this assumes that procedures are followed consistently without room for lack of compliance. Furthermore, air filtration has shown to reduce the risk of PRRSV outbreaks, which has rapidly been adopted in the US swine industry. In addition to filtration, producers and veterinarians have also implemented additional biosecurity measures in these herds, which may in the future in fact provide us with an opportunity to assess how much of the risk reduction is due to filters and how much is due to consistently complying with basic biosecurity measures. An important factor that likely partially explain contradictory results showed by different research studies is related to PRRSV strain and species differences on their capability to be shed nasally, become airborne, and remain infectious for a transmission event. This should be considered in the design of future studies that aim to elucidate the importance of airborne PRRSV transmission. Other factors such as immunity, herd size and dynamics, meteorological conditions, and regional density may also help explain differences in the likelihood of aerosol transmission and the conflicting reports available in the literature. Considering the peer-reviewed publications summarized herein, one can conclude that airborne transmission of PRRSV is possible; but the probability over long distances appears to be relatively low. Further studies are required to better understand whether airborne transmission is a frequent event and under what conditions this occurs. New sampling methods, epidemiological models, and diagnostic capabilities may be needed to further advance the knowledge of PRRSV airborne transmission.

## Figures and Tables

**Table 1 viruses-11-00712-t001:** Compiled list and main characteristics of peer-reviewed publications that directly aimed to assess airborne transmission of porcine reproductive and respiratory syndrome virus (PRRSV) within and/or between swine herds.

Authors ^1^	Year Published	Type of Study ^2^	PRRSV Strain(s) Used ^3^	Location ^4^	Transmission Distance Range Examined (Details)	Number Animals Used ^5^	Number of Replicates Experiments	Duration of Exposure	Airborne Transmission Suggested
Wills et al. [19]	1997	Experimental	ATCC VR-2402 (PRRSV-2)	USA	46–102 centimeters (within room)	65	5	31 days	No
Torremorell et al. [27]	1997	Experimental	MN-1b and VR-2332 (PRRSV-2)	Minnesota, USA	1 m (connecting pipe)	46	2	Up to 7 weeks	Potential
Otake et al. [33]	2002	Semi-Experimental	MN-30100 (PRRSV-2)	USA	1–80 m (within room, outside trailer, nearby facilities)	210	1	21 days	No
Trincado et al. [34]	2004	Semi-Experimental	MN-30100 (PRRSV-2)	USA	15–30 m (plastic tube connector, nearby facility)	165	1	21 days	No
Kristensen et al. [31]	2004	Experimental	PRRSV-1	Denmark	1 m (connecting pipes)	286	3	28–35 days	Yes
Dee et al. [32]	2005	Experimental	MN-30100 (PRRSV-2)	USA	1­–150 m (connecting pipe)	6	5	3 hours	
Fano et al. [35]	2005	Semi-Experimental	MN 30,100 (PRRSV-2)	Minnesota, USA	1–6 m (within room, connecting pipes, outside trailer)	63	1	7 days	No
Cho et al. [5]	2007	Experimental	MN-184 and MN-30100 (PRRSV-2)	Minnesota, USA	1–2 m	46	2	5 days	Dependent on genotype
Pitkin et al. [36]	2009	Semi-Experimental	MN-184 (PRRSV-2)	Minnesota, USA	4–120 m (nearby facilities)	1340	26	14 days	Yes
Dee et al. [28]	2009	Semi-Experimental	MN-184 (PRRSV-2)	Minnesota, USA	1.7–4.7 km (air samples followed by bioassay)	304	1	50 days	Potential
Otake et al. [39]	2010	Semi-Experimental	RFLPs 1-8-4, 1-18-2, 1-26-2 (PRRSV-2)	Minnesota, USA	1.4–10.2 km (air samples followed by bioassay)	314	1	21 days	Yes

^1^ Complete reference list with all authors and journal details can be found under the “References” section; ^2^ Experimental studies: studies controlled under highly controlled conditions (e.g., using pipes or connectors between populations); semi-experimental studies: studies controlled under controlled conditions but using a large sample size and attempting to mimic commercial conditions; field studies: studies conducted under commercial conditions and using “real” life data; ^3^ PRRSV-1 refers to the European PRRSV species, PRRSV-2 refers to the North American PRRSV species. Details in virus type was included for instances in which they were detailed by the authors.; ^4^ Geographical location where study was conducted. The state/province in which study was carried was noted for cases in which it was provided in the publication; ^5^ Approximate total number of animals used in study; calculations were based on described methodology and included challenged animals, sentinels and controls; for all replicates (as applicable).

**Table 2 viruses-11-00712-t002:** Summary of peer-reviewed publications of field-based studies assessing area spread or airborne transmission of porcine reproductive and respiratory syndrome virus (PRRSV).

Reference ^1^	Year	Country ^2^	Virus Type Studied ^3^	Airborne Transmission or Area Spread Main Route Suggested
Edwards et al. [41]	1992	United Kingdom	PRRSV-1	Yes
Goldberg et al. [43]	2000	US	PRRSV-2	Yes
Mortensen et al. [22]	2002	Denmark	PRRSV-2 (vaccine)	Yes
Mondaca-Fernandez et al. [44]	2006	US	PRRSV-2	No
Dee et al. [51]	2010	US	PRRSV-2	Potential
Spronk et al. [52]	2010	US	PRRSV-2	Potential
Linhares et al. [40]	2012	US	PRRSV-2	Yes
Alonso et al. [45]	2013	US	PRRSV-2	Yes
Rosendal et al. [48]	2014	Canada	PRRSV-2	No
Brito et al [48]	2014	US	PRRSV-2	Yes
Arruda et al. [46]	2015	Canada	PRRSV-2	Potential
Arruda et al. [47]	2017	Canada	PRRSV-2	Dependent on genotype
Arruda et al. [53]	2018	US	PRRSV-2/1	No

^1^ Complete reference list with all authors and journal details can be found under the “References” section; ^2^ Geographical location where study was conducted. The state/province in which study was carried was noted for cases in which it was provided in the publication; ^3^ PRRSV-1 refers to the European PRRSV species, PRRSV-2 refers to the North American PRRSV species. Details in virus type was included for instances in which they were detailed by the authors.

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
