# Peer review of "Aerosol Detection and Transmission of Porcine Reproductive and Respiratory Syndrome Virus (PRRSV): What Is the Evidence, and What Are the Knowledge Gaps?"

_viruses, 2019, doi:10.3390/v11080712_

Round 1
Reviewer 1 Report
This review on PRRSV aerosol transmission is well conducted and gathers a bunch of publications and information much needed in the field.
My only major remark is that I would highlight a little more the role of PRRSV strains, in their differential capacities to be nasally shed, to survive to an airborne event and finally their differential infectious capacities, 3 viral particles properties that must strongly determine the contradictory results obtained by different teams using different viral strains. Indeed, this is well illustrated by Cho et al. article. This should be highlighted especially in the conclusion, to invite scientist to proceed to strains comparative studies monitoring these parameters.
In line with this remark I would encourage the authors to more systematically mention the strain used in each study (for instance the information is missing lines 226, 238, 285, 359).
Minor remarks:
Line 114: PRRSV is an enveloped virus.
Line 127: Virus strain cannot 'alter' the virus ability to become airborne since virus strain is intrinsic to the virus particle itself. The capacity to be airborne varies according to the virus strain.
Line 179 and following: It would be important here to mention more explicitly the clear link between viremia and virus shedding with PRRSV detection in aerosol.
Line 262: This study, which is one of the authors own work, is quite complex, and I would suggest to clearly synthesis the main conditions, results and conclusions. For instance, can one considers that one is looking at M Hyo/PRRSV coinfections? Are there any clear conclusions that one can draw from the 2 variant PRRSV (1-18-2 and 1-26-2) introduced, how would it be more simple not to mention this over complexity of the study?
Author Response
Answers to Reviewers
Reviewer 1
This review on PRRSV aerosol transmission is well conducted and gathers a bunch of publications and information much needed in the field.
My only major remark is that I would highlight a little more the role of PRRSV strains, in their differential capacities to be nasally shed, to survive to an airborne event and finally their differential infectious capacities, 3 viral particles properties that must strongly determine the contradictory results obtained by different teams using different viral strains. Indeed, this is well illustrated by Cho et al. article. This should be highlighted especially in the conclusion, to invite scientist to proceed to strains comparative studies monitoring these parameters.
AA: We have inserted a sentence emphasizing this point in the Conclusion section; lines 527-531.
In line with this remark I would encourage the authors to more systematically mention the strain used in each study (for instance the information is missing lines 226, 238, 285, 359).
AA: These have been added on the text throughout the manuscript for all cases they were available on the original publication (e.g. for Stein et al, only ‘PRRSV-1’ was used with no further strain specification).
Minor remarks:
Line 114: PRRSV is an enveloped virus.
AA: Modified.
Line 127: Virus strain cannot 'alter' the virus ability to become airborne since virus strain is intrinsic to the virus particle itself. The capacity to be airborne varies according to the virus strain.
AA: We modified the wording on lines 130-132.
Line 179 and following: It would be important here to mention more explicitly the clear link between viremia and virus shedding with PRRSV detection in aerosol.
AA: This paragraph has been rewritten to add details on the strains used on this study as highlighted previously, as well as to add data on virus shedding results (lines 177-193).
Line 262: This study, which is one of the authors own work, is quite complex, and I would suggest to clearly synthesis the main conditions, results and conclusions. For instance, can one considers that one is looking at M Hyo/PRRSV coinfections? Are there any clear conclusions that one can draw from the 2 variant PRRSV (1-18-2 and 1-26-2) introduced, how would it be more simple not to mention this over complexity of the study?
AA: This paragraph was rewritten to simplify and clarify information from this complex study; lines 270-286.
Reviewer 2 Report
The manuscript is an extensive review of studies exploring an airborne transmission in case of porcine reproductive and respiratory syndrome virus. Authors thoroughly present the history of research and the latest developments within this field, as well as technical challenges in performing such analyses. Additionally, several most important knowledge gaps, still requiring further studies, were identified.
Considering conflicting results obtained by different authors, and difficulties in interpretation of different types of studies (experimental, semi-experimental and field), this review is a meaningful contribution to summarizing and analysis of current knowledge and indicating future research directions.
Some minor comments:
- Authors use expressions “PRRSV-1 or PRRSV-2 strain” to describe North American and European-type PRRSV (e.g. lines 241-242). According to the newest classification of ICTV, PRRSV-1 and PRRSV-2 are two separate species, so in fact, the paper describes research performed on two distinct viruses. This fact should be clearly stated and discussed as a possible source of observed diverse results. Also, referring to the “North American and European” nomenclature can be misleading without explanation, that the occurrence of both groups is worldwide nowadays.
- An Additional column in Table 1 including summary information on obtained results would facilitate the analysis of presented results (similarly as the last column in Table 2).
- The authors describe the scenario of airborne transmission as more probable when introducing the virus into the naïve population. However, there are also negative examples of such situation, when no airborne transmission was detected after the introduction of PRRSV into the negative countries - Switzerland and Sweden (Nathues et al. 2016, Carlsson et al. 2009), which were not mentioned here.
- In the Conclusions section authors discuss PRRS airborne transmission as an important area of research as well as influencing factors, but I miss some summarizing conclusion. Can authors comment on the overall dataset? Considering currently available data, the airborne transmission of PRRSV is possible but the probability is low, especially over the long distances. This may determine airborne as a more important route for within-herd than between-herds transmission.
Author Response
Reviewer 2
The manuscript is an extensive review of studies exploring an airborne transmission in case of porcine reproductive and respiratory syndrome virus. Authors thoroughly present the history of research and the latest developments within this field, as well as technical challenges in performing such analyses. Additionally, several most important knowledge gaps, still requiring further studies, were identified.
Considering conflicting results obtained by different authors, and difficulties in interpretation of different types of studies (experimental, semi-experimental and field), this review is a meaningful contribution to summarizing and analysis of current knowledge and indicating future research directions.
Some minor comments:
- Authors use expressions “PRRSV-1 or PRRSV-2 strain” to describe North American and European-type PRRSV (e.g. lines 241-242). According to the newest classification of ICTV, PRRSV-1 and PRRSV-2 are two separate species, so in fact, the paper describes research performed on two distinct viruses. This fact should be clearly stated and discussed as a possible source of observed diverse results. Also, referring to the “North American and European” nomenclature can be misleading without explanation, that the occurrence of both groups is worldwide nowadays.
AA: Information regarding the 2 different species as well as recognition of this as an explanation for contradictory study results have been added to the manuscript on lines 83-87. As studies using PRRSV-1 species are scarce, this was also mentioned in the manuscript (lines 466-469). Lastly, because conclusions regarding aerosol transmissibility differ even according to strains within PRRSV-2; the issues regarding species and strain differences have both been recognized as potential source for differing research results under the “Conclusion” section (as well as per request of the other reviewer), lines 527-531.
- An Additional column in Table 1 including summary information on obtained results would facilitate the analysis of presented results (similarly as the last column in Table 2).
AA: This column has been added to Table 1.
- The authors describe the scenario of airborne transmission as more probable when introducing the virus into the naïve population. However, there are also negative examples of such situation, when no airborne transmission was detected after the introduction of PRRSV into the negative countries - Switzerland and Sweden (Nathues et al. 2016, Carlsson et al. 2009), which were not mentioned here.
AA: We appreciate the references and we have included those as well as a comment that area spread is complex and likely context/ area/ environment-dependent (besides host and agent-dependent) in our Discussion, lines 498-502.
- In the Conclusions section authors discuss PRRS airborne transmission as an important area of research as well as influencing factors, but I miss some summarizing conclusion. Can authors comment on the overall dataset? Considering currently available data, the airborne transmission of PRRSV is possible but the probability is low, especially over the long distances. This may determine airborne as a more important route for within-herd than between-herds transmission.
AA: We have added a sentence on overall conclusion using the summarized papers, lines 534-536.